# Heterotypic Stressors Unmask Behavioral Influences of PMAT Deficiency in Mice

**DOI:** 10.3390/ijms242216494

**Published:** 2023-11-18

**Authors:** Brady L. Weber, Marissa M. Nicodemus, Allianna K. Hite, Isabella R. Spalding, Jasmin N. Beaver, Lauren R. Scrimshaw, Sarah K. Kassis, Julie M. Reichert, Matthew T. Ford, Cameron N. Russell, Elayna M. Hallal, T. Lee Gilman

**Affiliations:** Department of Psychological Sciences, Brain Health Research Institute, Healthy Communities Research Institute, Kent State University, Kent, OH 44240, USA

**Keywords:** stress, mice, fear conditioning, swim, corticosterone, sex differences, behavior

## Abstract

Certain life stressors having enduring physiological and behavioral consequences, in part by eliciting dramatic signaling shifts in monoamine neurotransmitters. High monoamine levels can overwhelm selective transporters like the serotonin transporter. This is when polyspecific transporters like plasma membrane monoamine transporter (PMAT, *Slc29a4*) are hypothesized to contribute most to monoaminergic signaling regulation. Here, we employed two distinct counterbalanced stressors—fear conditioning and swim stress—in mice to systematically determine how reductions in PMAT function affect heterotypic stressor responsivity. We hypothesized that male heterozygotes would exhibit augmented stressor responses relative to female heterozygotes. Decreased PMAT function enhanced context fear expression, an effect unexpectedly obscured by a sham stress condition. Impaired cued fear extinction retention and enhanced context fear expression in males were conversely unmasked by a sham swim condition. Abrogated corticosterone levels in male heterozygotes that underwent swim stress after context fear conditioning did not map onto any measured behaviors. In sum, male heterozygous mouse fear behaviors proved malleable in response to preceding stressor or sham stress exposure. Combined, these data indicate that reduced male PMAT function elicits a form of stress-responsive plasticity. Future studies should assess how PMAT is differentially affected across sexes and identify downstream consequences of the stress-shifted corticosterone dynamics.

## 1. Introduction

Under stressful environmental conditions, signaling patterns of monoamine neurotransmitters like dopamine and serotonin change dramatically [1,2,3,4]. The duration and magnitude of monoamine neurotransmitter signaling are regulated, in part, by transporter-mediated uptake. Monoamine neurotransmitter transporters are broadly categorized into two groups based upon their transport capacity (high or low) and their selectivity for substrates (see reviews [5,6,7]). Historically, the most well-studied monoamine transporters are those that have relatively high substrate selectivity and lower transport capacity. These include norepinephrine (*Slc6a2*), dopamine (*Slc6a3*), and serotonin (*Slc6a4*) transporters. In contrast, monoamine transporters that have higher capacity for substrate transport but are less selective about the substrates they transport include the organic cation transporters (*Slc22a1*, *Slc22a2*, *Slc22a3*; also known as OCT1, OCT2, and OCT3, respectively) and plasma membrane monoamine transporter (PMAT, *Slc29a4*). PMAT preferentially transports dopamine and serotonin over other monoamine neurotransmitters like norepinephrine or histamine [8] (see review [9]). Thus, PMAT function likely impacts dopamine and serotonin signaling, particularly during high signaling periods like stressful environmental conditions.

Previous studies in mice show that constitutive reductions in, or loss of, PMAT function affect behavioral responses to stressful environmental conditions, such as a swim stress [10] or tail suspension test [11]. Moreover, these behavioral responses were sex-specific. Mice constitutively lacking OCT2 exhibit augmented behavioral responses to both acute (both sexes used, but sex differences not analyzed; [12]) and chronic (males only; [13]) stressors. In contrast, male mice constitutively deficient in, or lacking, OCT3 exhibited no changes in the resident–intruder test nor in Morris water maze performance [14]. Findings with the latter test suggest that OCT3 does not affect spatial learning or memory processes, whereas the outcomes with both tests indicate that OCT3 is not involved in behavioral responses to stressors (aggressive encounters or water immersion).

Surprisingly few evaluations have examined how PMAT or the OCTs contribute to learning and memory processes. Moreover, no studies have assessed fear conditioning either in PMAT or OCT knockout mice or after administration of the broad OCT + PMAT inhibitor, decynium-22. Beyond the Morris water maze study mentioned earlier, a couple of groups have assessed conditioned place preference (CPP)—a form of classical conditioning—in OCT3 or PMAT knockout mice. Gautron’s group observed no influence of OCT3 knockout on amphetamine CPP (sex(es) not stated; [15]), whereas Daws’ group reported that males (but not females) lacking OCT3 had attenuated amphetamine CPP [16]. The latter group used a dose half that of the former group, which may explain some of the discrepancies between findings. In contrast, Daws’ group found that females (but not males) deficient in PMAT exhibited enhanced amphetamine CPP. Thus, some evidence exists that sex-specific influences of these understudied transporters could influence learning and memory processes. Still, because amphetamine affects monoamine signaling, disentangling the effects of PMAT/OCT deficiency upon responses to amphetamine from those upon learning and memory is difficult to do.

We evaluated how PMAT deficiency influences classical conditioning in the absence of any drug exposure in the present investigation. We accomplished this using both contextual and cued fear conditioning paradigms in conjunction with exposure to a second, different form of stressor—swim stress. Contextual and cued fear conditioning preferentially engage activity within the dorsal hippocampus and amygdala [17,18,19] (see reviews [20,21,22]), whereas swim stress predominantly increases hypothalamus and amygdala activity [23,24,25]. Here, we assessed directional influences of these two different stressor formats—context/cued fear conditioning before/after swim stress—by evaluating stress-responsive behaviors specific to each paradigm (freezing; swimming, climbing, immobility; respectively) and circulating blood corticosterone levels as a proxy index of stress persistence. Because we have previously observed sex-specific stressor responses in PMAT-deficient mice [10,11], these studies were likewise performed in mice of both biological sexes. Finally, we intentionally used only wild-type (+/+) and heterozygous (+/−) PMAT mice, given potential translational relevance to humans with functional reductions in PMAT resulting from common polymorphisms [26,27,28,29].

We hypothesized that attenuated PMAT function in heterozygous mice would enhance behavioral responses to both initial and secondary stressors due to reduced clearance of elevated dopamine and serotonin. Further, we hypothesized that male heterozygotes would exhibit augmented behavioral and physiological stressor responses relative to females, given previous indications of such in male PMAT-deficient mice [11], plus literature evidence suggesting overall sex differences in stressor responsivity in mice [30,31].

## 2. Results

### 2.1. Fear Behavior

#### 2.1.1. Phase 1

Because this is the first report of fear conditioning in PMAT-deficient mice, we began with Phase 1 experiments (Figure 1), which involved first performing fear conditioning followed four weeks thereafter by swim stress. This allowed for the initial identification of any influences of PMAT deficiency upon fear processing, independent of prior swim stress exposure. As expected, there were no significant interactions with swim stress condition, nor any main effects of swim, for Phase 1 cued (Appendix A) and Phase 1 context (Appendix A) experiments across sexes. Consequently, graphed data were collapsed across swim condition to focus upon the effects of time and genotype, and the interactions thereof as applicable.

##### Phase 1 Cued Females

In Phase 1 cued females, there was an expected main effect of time for training (F_(3.122,109.266)_ = 183.4, *p* < 0.001, partial η^2^ = 0.840), extinction retention testing (F_(9.774,342.097)_ = 8.024, *p* < 0.001, partial η^2^ = 0.186), context fear expression (F_(6.926,242.403)_ = 5.297, *p* < 0.001, partial η^2^ = 0.131), and cued fear renewal (F_(3.712,129.923)_ = 12.40, *p* < 0.001, partial η^2^ = 0.262) (Appendix A; Figure 2A,C–E). While there were no interactions with, nor main effects of, genotype for any of these, there was a significant time × genotype interaction (F_(9.728,340.487)_ = 3.865, *p* < 0.001, partial η^2^ = 0.099) in Phase 1 cued females for cued expression testing and extinction training (Appendix A; Figure 2B). Post hoc testing reflects that female heterozygous mice exhibited a temporally distinct pattern of cued fear expression and cued fear extinction from wild-type females at multiple time points (Figure 2B). This appears to be the result of female heterozygotes displaying delayed cued fear extinction relative to female wild types.

##### Phase 1 Cued Males

Unlike females that underwent Phase 1 cued procedures, Phase 1 cued males had no interactions with, or main effects of, genotype at any stage (Appendix A; Figure 2F–J). The anticipated main effect of time was present for all stages in Phase 1 cued males (training, F_(3.156,170.433)_ = 205.7, *p* < 0.001, partial η^2^ = 0.792; cued expression testing and extinction training, F_(10.594,572.057)_ = 9.786, *p* < 0.001, partial η^2^ = 0.153; extinction retention testing, F_(10.125,536.64)_ = 11.09, *p* < 0.001, partial η^2^ = 0.173; context fear expression, F_(10.889,533.558)_ = 11.58, *p* < 0.001, partial η^2^ = 0.191; cued fear renewal, F_(3.82,187.198)_ = 11.17, *p* < 0.001, partial η^2^ = 0.186) (Appendix A; Figure 2F–J). Combined, these data indicate that PMAT deficiency was largely without consequence on cued fear processing in males, whereas it had a modest impact on cued extinction learning in females.

##### Phase 1 Context Females

Females assigned to the Phase 1 context condition exhibited no interactions of time × genotype, and no main effect of genotype, but the expected main effect of time during context fear training (F_(3.626,123.285)_ = 125.3, *p* < 0.001, partial η^2^ = 0.787) (Appendix A; Figure 3A). Similarly, testing of context fear expression in these females revealed a significant main effect of time (F_(9.138,310.686)_ = 8.874, *p* < 0.001, partial η^2^ = 0.207) (Appendix A; Figure 3B). A non-significant trend of genotype (F_(1,34)_ = 2.936, *p* = 0.096, partial η^2^ = 0.079) was noted for females during context fear testing, but pairwise comparisons did not indicate any select time points that differed significantly across genotype.

##### Phase 1 Context Males

Males that underwent context fear conditioning were similar to females in that only a main effect of time was detected for training (F_(3.226,125.803)_ = 12.92, *p* < 0.001, partial η^2^ = 0.249) (Appendix A; Figure 3C). Unlike females, however, context fear testing in males showed main effects of both genotype (F_(1,39)_ = 4.555, *p* = 0.039, partial η^2^ = 0.105) and time (F_(9.546,372.282)_ = 13.94, *p* < 0.001, partial η^2^ = 0.263) (Appendix A; Figure 3D). Heterozygous males exhibited increased context fear expression compared to wild-type males (Figure 3D; Appendix A), suggesting that typical PMAT function could selectively suppress the expression of context fear in males.

#### 2.1.2. Phase 2

After determining how reductions in PMAT function impact cued and context fear processing in the absence of any preceding stressors, we next evaluated how heterotypic stressor exposure interacted with PMAT function and sex. To do this, mice underwent cued or context fear conditioning procedures identical to those in Phase 1, except these procedures occurred four weeks after a swim stressor.

##### Phase 2 Cued Females

Repeated-measures ANOVAs of females in the Phase 2 cued condition indicated that there were no three-way time × genotype × swim interactions in any of the five stages: training, cued expression testing and extinction training, extinction retention testing, context fear expression, or cued fear renewal (Appendix A; Figure 4). There were also no time × genotype interactions across these five stages, nor were any main effects of genotype or swim detected (Appendix A; Figure 4). The first four stages all had the expected main effect of time (training, F_(3.605,147.811)_ = 159.1, *p* < 0.001, partial η^2^ = 0.795; cued expression testing and extinction training, F_(8.741,358.393)_ = 6.623, *p* < 0.001, partial η^2^ = 0.139; extinction retention testing, F_(8.864,310.252)_ = 11.15, *p* < 0.001, partial η^2^ = 0.242; context fear expression, F_(10.762,441.257)_ = 5.923, *p* < 0.001, partial η^2^ = 0.126) (Appendix A; Figure 4A–D,F–I). Cued fear renewal was the only stage with a significant time × swim interaction (F_(3.57,139.219)_ = 3.592, *p* < 0.001, partial η^2^ = 0.0.84) (Appendix A; Figure 4E,J). With pairwise comparisons, we determined that this appeared to be driven by reduced extinction of cued fear renewal over time in mice that previously underwent a swim stressor (Figure 4J). This was most prominent in heterozygous females, reaching significance for cued fear in response to the final tone between heterozygous females that had a swim stressor compared to heterozygous females that did not undergo swim stress (Figure 4J). This partially mirrors the apparent impaired cued fear extinction learning exhibited by Phase 1 cued females on testing Day 2.

##### Phase 2 Cued Males

Phase 2 cued males likewise showed no three-way time × genotype × swim interactions, nor any two-way interactions across all five fear behavior testing stages (Appendix A; Figure 5). No main effect of swim was detected at any stage, whereas significant main effects of time occurred for all stages (training, F_(2.700,97.197)_ = 123.1, *p* < 0.001, partial η^2^ = 0.774; cued expression testing and extinction training, F_(8.933,267.996)_ = 10.55, *p* < 0.001, partial η^2^ = 0.260; extinction retention testing, F_(7.79,296.005)_ = 8.583, *p* < 0.001, partial η^2^ = 0.184; context fear expression, F_(9.633,356.405)_ = 5.218, *p* < 0.001, partial η^2^ = 0.124; cued fear renewal, F_(3.835,130.383)_ = 24.30, *p* < 0.001, partial η^2^ = 0.417) (Appendix A; Figure 5). Significant main effects of genotype were found for Phase 2 cued males for both extinction retention testing (F_(1,38)_ = 6.914, *p* = 0.012, partial η^2^ = 0.154) (Figure 5C,H) and context fear expression (F_(1,37)_ = 4.175, *p* = 0.048, partial η^2^ = 0.101) (Figure 5D,I), the latter reflecting what was found for Phase 1 context males (Figure 3D), but not context testing in Phase 1 cued males (Figure 2I). Pairwise comparisons highlight that, within the no-swim condition, heterozygous males in Phase 2 cued extinction retention testing exhibited impaired retention of extinction training relative to wild types (Figure 5C). Put another way, heterozygous males that did not first undergo swim stress displayed persistent fear in response to tones that no longer predicted foot shock, indicating that they did not retain the extinction learning they had undergone on testing Day 2. Further, pairwise comparisons suggest that the genotype effect in context fear expression appears to be mostly driven by males in the no-swim condition (Figure 5D), despite the absence of any significant swim effects or interactions. These Phase 2 findings provide further support for a sex-dependent role of intact PMAT function attenuating expression of context fear and additionally suggest that PMAT might typically function to facilitate retention of cued extinction in males.

##### Phase 2 Context Females

No three- or two-way interactions were found for females in the Phase 2 context condition during either training or testing, and the only main effects were those of time (training, F_(3.029,102.984)_ = 76.52, *p* < 0.001, partial η^2^ = 0.692; testing, F_(9.209,276.265)_ = 8.773, *p* < 0.001, partial η^2^ = 0.226) (Appendix A, Figure 6).

##### Phase 2 Context Males

Males in the Phase 2 context similarly had no significant three-way interactions for training or testing. No two-way interactions were found for training. Training did exhibit the anticipated main effect of time (F_(3.284,108.38)_ = 66.40, *p* < 0.001, partial η^2^ = 0.668). Neither time × genotype nor time × swim interactions were significant for testing (Appendix A; Figure 7). A non-significant trend was observed for swim × genotype (F_(1,33)_ = 3.400, *p* = 0.074, partial η^2^ = 0.093) during context fear testing in males, as was a significant main effect of time (F_(8.625,284.634)_ = 14.87, *p* < 0.001, partial η^2^ = 0.311) (Appendix A; Figure 7B,D). Pairwise comparisons suggest that context fear expression was lowered by previous swim stress exposure in wild types, while swim stress in heterozygous males increased their context fear expression (Figure 7B,D). These changes in context fear expression across genotypes from an earlier stressor in males appear to be the inverse of what is observed for context fear expression in Phase 2 males that underwent cued fear conditioning (Figure 5). In other words, prior swim stress hid genotype differences in context fear in Phase 2 cued males, but prior swim stress made more apparent genotype differences in context fear in Phase 2 context males. Though complex in directionality and circumstance, overall these fear behavior data indicate that reductions in PMAT function result in more prominent behavioral effects in males versus females. Further, the present findings suggest that the form of fear conditioning and stressor history can sex-specifically enhance or mask the influence of diminished PMAT function.

### 2.2. Serum Corticosterone

Blood was collected from all mice 2 h after their last test to measure serum corticosterone levels. For Phase 1 mice, this was 2 h after swim stress; for Phase 2 mice, this was 2 h after the context testing and cued fear renewal test. This 2 h time point was to capture the descending limb of the corticosterone curve to assess how elevated corticosterone levels remained after the established 30 min peak post-stressor [32,33,34,35,36]. This 2 h time point was the focus here because no sex nor genotype differences were detected in PMAT mice when serum corticosterone was measured 30 min following a single acute swim stressor (Appendix A). All serum corticosterone levels were log-transformed (hereafter referred to as “cort”) to ensure normal distribution, as previously reported [37,38,39,40].

#### 2.2.1. Phase 1 Cued

Analyzing Phase 1 cued cort levels via three-way ANOVA indicated no significant three- or two-way interactions (Appendix A; Figure 8). The only significant main effect was one of sex (F_(1,82)_ = 17.77, *p* < 0.001, partial η^2^ = 0.178)) (Appendix A). Post hoc testing indicated that male wild-type mice in the no-swim condition exhibited significantly lower cort levels than female wild-type no-swim mice (Figure 8A,C). Conversely, in the swim condition, male heterozygous mice had lower cort levels than female heterozygotes (Figure 8A,C). Such sex differences in descending cort levels have been reported previously [36,41,42,43,44] (see review [45]).

#### 2.2.2. Phase 1 Context

Unlike Phase 1 cued cort levels, there was a significant three-way interaction of genotype × sex × swim (F_(1,72)_ = 6.029, *p* = 0.016, partial η^2^ = 0.077) (Appendix A, Figure 8). Pairwise comparisons showed that male heterozygotes in the swim condition had significantly lower cort levels than male wild types in the same condition (Figure 8D). Male wild types that underwent swim stress also had significantly higher cort levels than male wild types not exposed to swim stress (Figure 8D). Several sex differences were also detected; specifically, cort levels for males were lower than females for all swim-genotype combinations except swam wild types (Figure 8B,D). These data could indicate that prior context fear conditioning augments cort elevations to subsequent acute stressors in wild-type males but that reduced PMAT function dampens this response.

#### 2.2.3. Phase 2 Cued

Similar to Phase 1 cued, no significant three- nor two-way interactions were detected in cort levels for Phase 2 cued (Appendix A). As with Phase 1 cued, the only significant main effect in Phase 2 cued was of sex (F_(1,79)_ = 17.40, *p* < 0.001, partial η^2^ = 0.180) (Appendix A; Figure 9). Unlike Phase 1 cued cort levels though, pairwise comparisons indicated no specific differences between individual groups (Figure 9A,C). The absence of any swim effect for Phase 2 cued mice indicates that an acute swim stress 4 weeks prior to undergoing cued fear conditioning was not impactful enough to alter cort responses in the long term.

#### 2.2.4. Phase 2 Context

No significant three-way interaction of genotype × sex × swim (Appendix A) was detected. Though genotype × sex and sex × swim interactions were not significant, a genotype × swim interaction was significant for Phase 2 context (F_(1,63)_ = 4.377, *p* = 0.040, partial η^2^ = 0.065) (Appendix A). As with Phase 2 cued though, post hoc testing indicated no specific differences between any two groups (Figure 9B,D). Considering these data along with the cort levels of Phase 2 cued mice, it is possible that the prolonged testing for cued fear conditioning obscured any lasting cort regulatory changes. In contrast, context fear conditioning’s more concise timeline could have facilitated a glimpse into the impact of PMAT genotype upon cort levels in Phase 1 following an earlier acute stressor.

### 2.3. Swim Stress

For both Phases 1 and 2, only those mice assigned to the swim stress condition underwent a 6 min swim stress. Mice assigned to the no-swim condition were transported to the same room and treated the same as mice that underwent swim (i.e., placed in clean cages half-on heating pads) but were not swam. All swim stresses were video-recorded, then later hand-scored offline by two blinded observers to quantify swimming, climbing, and immobility behaviors for subsequent analyses. Additionally, latency to the first bout of immobility (i.e., “latency”) and the number of fecal boli (Appendix A) were analyzed.

#### 2.3.1. Phase 1 Cued

Mice that underwent cued fear conditioning 4 weeks prior to swim stress exhibited no significant interactions between genotype × sex, nor any main effects of either genotype or sex (Appendix A). Accordingly, no significant post hoc tests were observed either (Figure 10A,C). This aligns with the absence of any cort level differences detected in Phase 1 cued mice (Figure 8A,C).

#### 2.3.2. Phase 1 Context

Mice that were swam 4 weeks after undergoing context fear conditioning had a significant genotype × sex interaction in swimming behaviors (F_(1,36)_ = 5.572, *p* = 0.024, partial η^2^ = 0.134) (Appendix A). Post hoc testing indicated that male heterozygotes displayed significantly more swimming behavior than female heterozygotes (Figure 10D). Despite the absence of any genotype-specific behavioral changes observed in swim behaviors, cort measurements indicate that heterotypic stressor exposure across the 4 week experimental timeframe did indeed have a physiological impact in males that appears to be moderated by PMAT deficiency (Figure 8B,D).

#### 2.3.3. Phase 2 Cued

In Phase 2, mice underwent a swim stress 4 weeks before fear conditioning. Mice that went through cued fear conditioning after swim stress had no significant differences in swimming (Appendix A). Though no significant interactions or main effects were detected for either immobility or climbing, both had non-significant trends for genotype (immobility, F_(1,42)_ = 3.221, *p* = 0.080, partial η^2^ = 0.071; climbing, F_(1,42)_ = 3.024, *p* = 0.089, partial η^2^ = 0.067) (Appendix A). Latency, while not having a significant genotype × sex interaction, did have a significant main effect of genotype (F_(1,42)_ = 4.679, *p* = 0.036, partial η^2^ = 0.100) (Appendix A; Figure 11A,C). Post hoc testing demonstrated that male heterozygotes that went through swim stress 4 weeks before cued fear conditioning displayed less immobility, and more climbing behavior, than male wild types that went through the same procedures (Figure 11C). The male-specific influences of reduced PMAT function on swim stress behavior reflect the largely consistent trend observed here, where male behavior and physiology were more affected than females.

#### 2.3.4. Phase 2 Context

Four weeks before going through Phase 2 context fear conditioning, mice were subjected to swim stress. Across all four measures of behavior, there was no significant genotype × sex interaction and no main effect of genotype, but there was a significant main effect of sex (swimming, F_(1,34)_ = 4.996, *p* = 0.032, partial η^2^ = 0.128; immobility, F_(1,34)_ = 10.09, *p* = 0.003, partial η^2^ = 0.229; climbing, F_(1,34)_ = 5.611, *p* = 0.024, partial η^2^ = 0.142; latency, F_(1,34)_ = 18.76, *p* < 0.001, partial η^2^ = 0.356) (Appendix A; Figure 11B,D). Post hoc tests emphasized significantly less time spent immobile, and an accompanying increase in latency, in males of both genotypes compared to females of the same genotype (Figure 11B,D). Why these sex differences were not also present in Phase 2 cued mice (Figure 11A,C) is not clear, though it could be attributable to the somewhat greater variability observed in Phase 2 cued females versus Phase 2 context females.

## 3. Discussion

### 3.1. Summary of Fear Behavior Findings

In addition to being the first assessment of how reduced PMAT function impacts classical conditioning to an aversive stimulus, the present findings additionally are an inaugural foray into systematically examining interactions between PMAT function and heterotypic stressor exposure. Previously, we noticed that sequential brief stressors in male PMAT-deficient mice altered behavior [11]. Consequently, we sought to explore this phenomenon in more depth, while simultaneously assessing how functional PMAT reductions affect fear-processing measures. Here, we observed that diminished PMAT expression shifts the time course of cued fear expression and cued extinction training in females, while augmenting the expression of context fear in males. Notably, PMAT function appeared to be without substantive impact upon acquisition of cued or context fear conditioning. This allows conclusions about how the function of PMAT moderates expression of fear to be made independent of any concerns regarding confounds of acquisition. Certainly, though, the contribution of PMAT to the consolidation of aversive memories remains to be clearly defined.

#### 3.1.1. Sex-Specific Impacts of PMAT Function on Fear Behavior

Previously, we and others have found sex-specific effects of PMAT deficiency on behavior [10,11,16] (see review [5]). Similar outcomes were found here. With the exception of cued fear expression and extinction training in females, the broad theme in the present findings was that females were largely unaffected by PMAT reductions or heterotypic stress exposure in their fear behavior. In this singular instance of fear behavior differences in female Phase 1 cued heterozygotes, it appears that reduced PMAT function in females impedes cued fear extinction, resulting in this interaction of genotype with time, the latter of which is required for the process of extinction to be detected.

In contrast to females, decreased PMAT function in males enhanced context fear expression prior to any preceding stress exposures (i.e., in Phase 1). This remained true for Phase 2 PMAT-deficient males that encountered a brief swim stressor 4 weeks before being trained in context fear conditioning. Combined, these data suggest an overall effect of reduced PMAT function on context fear expression in males, in contrast to the minor but significant interaction between reduced PMAT function and cued fear extinction within females. Put another way, functional PMAT in males appears to exert broader effects on *expression* of learned fear, whereas in females, influences of functional PMAT may be more restricted to specific instances of initial extinction learning.

Also counter to our hypothesis, Phase 2 cued PMAT-deficient males exhibited behavior that mostly mirrored that of behavior by pre-swim males in Phase 1 cued. In other words, swim stress exposure did not alter male mouse fear behaviors independent of genotype. Paradoxically, male Phase 2 context PMAT-deficient mice that experienced a sham procedure (no-swim) instead of a swim stress exhibited reduced context fear expression relative to their wild-type counterparts. Adding to the confusion were Phase 2 cued male heterozygous no-swim mice that exhibited enhanced context fear (like Phase 1 context males) preceded by impaired cued extinction retention.

#### 3.1.2. Hypotheses and Next Steps—Fear Behavior

While initially perplexing, we hypothesize that these data indicate that typical PMAT function might usually obscure enduring effects of modestly arousing experiences—such as those of no-swim mice being temporarily relocated for a sham swim. In contrast, exposure to an overt, albeit brief, swim stressor might induce enough of a neurophysiological perturbation to obscure this slight sensitivity present in heterozygotes. Moreover, the observed behavioral changes in fear expression and retention were mostly selective to males, suggesting a sex hormone component [46,47,48]. Indeed, recent studies are beginning to parse apart the molecular underpinnings of PMAT’s sex-specific functions [49,50]. Future experiments to test the long-term effects of arousing experiences at different intervals and durations could facilitate testing this hypothesis, as would gonadectomy experiments to determine if these are activational or organizational effects of sex hormones (or independent of sex hormones). To better characterize the nuances of cued fear expression and extinction learning in females, overtraining of females, combined with longer extinction training trials (e.g., presenting 30 tones instead of 15), could be useful.

### 3.2. Summary of Log-Transformed Corticosterone (Cort) Findings

Evaluating cort levels 2 h following the last behavioral test (swim, Phase 1; context testing and cued fear renewal, Phase 2) helped determine how PMAT deficiency interacted with prior stressor exposure to influence the return of cort levels to baseline. This was the focus here given previous evidence that reduced or ablated PMAT function had no impact on cort levels 30 min after an acute swim stressor (Appendix A); a time point accepted to be the peak of cort response to acute stress [32,33,34,35].

Overall, both Phase 1 cued and Phase 2 cued cort levels only exhibited significant sex differences, a known phenomenon where female mice typically have higher cort levels than males [41,51,52]. In contrast, Phase 1 context cort levels indicated that the return of cort levels to baseline following a swim stressor was influenced by the combination of swim, sex, and genotyping. Phase 2 context cort levels revealed a parallel interaction between swim and genotype. Thus, it could be that context fear is better suited for studying heterotypic stressor exposure than the more protracted cued fear conditioning paradigm, when used in combination with a swim stressor.

#### 3.2.1. Sex-Specific Impacts of PMAT Function and Stressor Exposure on Cort Levels

Expected sex effects [41,51,52] were observed in Phase 1 mice. Phase 1 cued mice exhibited no other effects, suggesting that their prior cued fear conditioning experiences did not manifest in cort level dynamics after mice were swam. In contrast to Phase 1 cued mice, Phase 1 context mice displayed the most robust interaction between the three variables of genotype, sex, and swim. Cort levels in male Phase 1 context wild-type swam mice had not returned to baseline, reflected in their non-swam counterparts, whereas male heterozygotes displayed cort levels similar to those of non-swam males independent of genotype. This indicates that the prior exposure of male heterozygotes to context fear conditioning may have either improved the mice’s ability to regulate HPA axis activation and return to baseline more quickly or that they exhibited a blunted cort response to the swim stressor.

Like Phase 1 cued, Phase 2 cued cort levels displayed the anticipated differences across sexes [41,51,52] but were without any other remarkable features. Because testing of fear expression, in the absence of any unconditioned stimulus, is by its nature less evocative of a stressor, the absence of prominent sex-, genotype- and swim-specific differences is not necessarily surprising for Phase 2 cued cort levels. Additionally, the five-day-long cued fear conditioning procedure may have led to some physiological habituation across all Phase 2 cued mice, dampening cort levels by the time the fifth day of testing arrived.

Only for Phase 2 context were sex differences not observed, and instead a genotype × swim interaction predominated. The loss of this sex difference in Phase 2 context could be due to the larger variability in cort levels but also could suggest a lingering effect of prior context fear conditioning on the physiological regulation of cort.

#### 3.2.2. Hypotheses and Next Steps—Cort Levels

While intriguing, these cort level differences (or lack thereof) do not map onto either fear or swim behavior, indicating that these physiological changes are likely exerting other influences that were not captured by the present study. Nonetheless, the significant interaction term detected for Phase 1 context suggests that the pairing of context fear conditioning and swim stress—in that order—might be more informative when optimized. Extended studies examining timelines of cort levels following an acute swim stressor that occurs 4 weeks after context fear conditioning would help answer our hypothesis that male heterozygous mice have cort levels that are returning to baseline faster, rather than exhibiting a blunted response. Alternatively, given the absence of concordance between cort levels and behavior here, future investigations could query what behaviors these cort levels do map onto, including risk assessment [53], depressive-like behavior [54], or social interaction [38] as possibilities.

### 3.3. Summary of Swim Behavior Findings

The swim behaviors exhibited by mice after (Phase 1) or before (Phase 2) fear conditioning were only modestly different across Phases. Phase 1 cued swim behavior revealed no effects or interactions of genotype and sex, whereas Phase 2 cued swim behaviors exhibited a genotype effect on latency plus non-significant trends for genotype in time spent immobile and climbing. Once again, Phase 1 context revealed sex-specific genotype effects, further supporting the combination of context fear testing followed by swim stress as a useful directional combination for uncovering the effects of heterotypic stress exposure and its interactions with genotypes related to stress responsivity. Phase 2 context, in contrast, was consistent only in a pervasive sex effect.

#### 3.3.1. Sex-Specific Impacts of PMAT Function on Swim Behavior

A swimming-specific genotype × sex interaction was found for Phase 1 context mice, where male heterozygotes swam more than female heterozygotes. Otherwise, prior exposure to either cued or context fear conditioning did not drastically alter swim behaviors. When Phase 2 swim behaviors were evaluated, 4 weeks before those mice would ever experience any form of fear conditioning, there was an unanticipated disparity in the overall patterns observed. Genotype effects were more prominent in Phase 2 cued mice, whereas sex effects dominated in Phase 2 context mice. Previously, we observed sex × genotype interactions during swim stress [10]. Because of the timing of the swim stressors in the current study though, differences between Phase 2 cued and Phase 2 context should not exist, particularly considering that the same two blinded observers scored all swim behaviors after all behavior testing had concluded. Thus, the findings noted for these should be interpreted with caution.

Mirroring findings for fear behavior and cort levels, swim behavior differences were largely specific to males. Phase 1 context swimming behavior moved in opposing directions between the sexes of heterozygous mice. Male Phase 2 cued heterozygotes showed less immobility and a corresponding increase in climbing behaviors, though the same was not observed for male Phase 2 context heterozygotes. Across genotypes in Phase 2 context, males exhibited less immobility and greater latencies to first immobility than females. Again though, this was inexplicably not replicated in Phase 2 cued mice. As with fear behavior, swim behaviors in both phases did not map onto cort levels, suggesting that PMAT function influences other physiological processes that drive swim behaviors, likely serotonin signaling [3,55,56], among others.

#### 3.3.2. Hypotheses and Next Steps—Swim Behavior

The differences in swim stress behavior in Phase 2 mice is disconcerting. Though we assigned mice to swim/no-swim and cued/context conditions using systematic randomization, and took care to have all swim behaviors scored by two blinded observers, we obtained data that did not replicate in our own hands. One adjustment we have considered, after hand-scoring of swim behaviors was completed and we discovered these perplexing outcomes, is to merely use the acute swim as a stressor and to not attribute much meaning to the behaviors that can be scored from it. Debate about the utility and interpretation of swim stress (also known as forced swim test) persists [57,58,59], with data suggesting that it is a better test of coping style, and it may best be suited for eliciting physiological stress responses (e.g., increasing circulating corticosterone). Indeed, both as a standalone inducer of acute cort increases and as a tool combined with preceding context fear conditioning to look at heterotypic stressor responses in a physiological manner, the swim stress has, at least in our hands, been consistently reliable for these purposes.

### 3.4. Limitations

The present study supports the contributions of PMAT function to behavioral and physiological heterotypic stress responses. Limitations of the study include the aforementioned differences in swim behavior prior to any conditioning exposure, a consequence of quantifying swim behaviors later than would have been optimal. Additionally, the discordance between cort changes and behavioral shifts indicates that alternative physiologic/behavior measures might have instead been better suited to detect concordance [38,53,54]. The absence of a complete time course of cort levels is another limitation but would have either resulted in a potentially confounding stress source or the use of more mice than we could ethically justify for the purpose of this investigation. In hindsight, focusing specifically on context fear conditioning and swim stress, in that order, and incorporating instead a behavioral measure of appetitive learning (e.g., lever pressing for a food reinforcer), might have provided better insight into the behavioral consequences of PMAT function upon heterotypic stress responsivity. Additionally, given the unreliability of commercial antibodies against PMAT, plus the absence of any selective PMAT inhibitors, it has been challenging to determine the level of PMAT protein expression—let alone PMAT function—in PMAT heterozygous mice. Consequently, the translatability of the present findings is hindered by this presently unobtainable information.

### 3.5. Overview

Nonetheless, the present findings provide important information upon which future experiments can be based to better focus efforts on understanding PMAT’s roles. Such studies should take a deeper look at learning and memory processes and explore both behavioral and molecular changes occurring from PMAT reductions and stressor encounters. Investigations employing orchidectomies could also provide insight into the organizational and/or activational interactions of sex hormones with PMAT function. This is only the second study to date to use classical conditioning In PMAT mice [16], and the first to employ fear conditioning, so more remains to be learned in this domain, including long-term memory, cue discrimination, and generalization, among other parameters. Moreover, expanding studies into evaluations of PMAT’s role in operant (rather than classical) conditioning procedures could be informative. And, as always with PMAT, the development of drugs that selectively inhibit this transporter would be a tremendous boon to understanding the functional influences of this protein.

## 4. Materials and Methods

### 4.1. Animals

Adult (≥90 days old) male and female PMAT-deficient mice maintained on a C57BL/6J background and bred in-house were used for all experiments. This line of mice was developed by Dr. Joanne Wang’s lab at the University of Washington [60]. Our PMAT-deficient colony is maintained in accordance with a material transfer agreement between the University of Washington and Kent State University. Males and females were run through all experiments separately; if both sexes were run on the same day, all males were always run before any females. All mice were group-housed (2–5 per cage) within the same sex on 7090 Teklad Sani-chip bedding (Envigo, East Millstone, NJ, USA). Mice had *ad libitum* access to LabDiet 5001 rodent laboratory chow (LabDiet, Brentwood, MO, USA) and drinking water. The vivarium was maintained at 22 ± 1 °C, on a 12:12 light:dark cycle, with lights on at 07:00. All procedures adhered to the National Research Council’s Guide for the Care and Use of Laboratory Animals, 8th Ed. [61], and were approved by the Kent State University Institutional Animal Care and Use Committee.

### 4.2. Genotyping

On postnatal day 21 (P21), mice were weaned, and 2 mm ear punches were collected for DNA extraction. Extensive details regarding buffer compositions, and procedures for DNA extraction, PCR (including primer sequences), and agarose gel electrophoresis, are published [10,60], including in an open-access journal [11].

### 4.3. Fear Conditioning

Mice underwent either contextual fear training or cued fear training (Figure 1). Regardless of the type of training, all mice were trained in ‘Context A’ in chambers made by Coulbourn Instruments (7 in D × 7 in W × 12 in H; Allentown, PA, USA). These chambers consisted of two opposite clear acrylic walls and two opposite aluminum panel walls. In Context A, the chamber contained a metal shock grid floor, had a blue dotted pattern hung behind one of the clear acrylic walls, was illuminated with visible light, and was cleaned with 70% ethanol as a scent cue. Sound-attenuating enclosures surrounded each separate chamber, and every chamber had a camera mounted at the top to record behavior. FreezeFrame (v. 5.201, Coulbourn Instruments) software was used to quantify freezing behavior in real time. Freezing behavior is defined as the absence of all movement except that required for breathing. Testing commenced 48 h after training for both contextual and cued fear conditioning paradigms. Mice were brought directly from the vivarium to the fear behavior room on every day of testing and training in a designated individual transport cage. Differences between context and cued fear conditioning paradigms are described below.

### 4.4. Cued Fear Conditioning

Following a 2 min baseline, training for cued fear involved five tone–shock pairings, with each 4 kHz, 30 s tone co-terminating with a 1 s, 0.8 mA scrambled mild foot shock. Inter-tone intervals (ITIs) of 90 s were used, and the entire training duration including baseline lasted 11 min. Percent freezing was measured for each 30 s period when a tone was played; this was graphed as cued fear training. Testing for cued fear began 48 h after training (Figure 1) and involved three stages, none of which included shocks. The first stage was for cued fear expression and cued fear extinction training; this included a 2 min baseline followed by fifteen 30 s, 4 kHz tone presentations separated by 30 s ITIs [62]. The second stage of testing began 48 h after the first testing stage. This had the exact same structure as the first stage of testing, but the purpose was to evaluate cued fear extinction retention, plus further cued fear extinction training. The first and second stages of testing occurred in ‘Context B’. Context B had a smooth acrylic floor, no pattern, was illuminated only with infrared light, and was cleaned with Windex^®^ (SC Johnson, Racine, WI, USA) as the scent cue. The third and final stage of testing occurred 24 h after the second stage. This third stage of testing occurred in Context A and contained two portions. First, behavior was observed in Context A for 10 min in the absence of any tones, to evaluate contextual fear expression and extinction. Then, the second portion began immediately at the 10 min point by presentation of five 30 s, 4 kHz tones separated by 30 s ITIs to assess cued fear renewal [63,64,65].

### 4.5. Context Fear Conditioning

Following a 2 min baseline, training for context fear involved pseudorandom delivery of five 1 s, 0.8 mA scrambled mild foot shocks delivered at 137, 186, 229, 285, and 324 s. The entire training duration including baseline lasted 6 min (Figure 1). Percent freezing was measured for each 30 s period—averaged across six 5 s bins—that followed each foot shock, starting with the first 5 s bin that did not include the foot shock. This was graphed as context fear training. Testing for context fear occurred 48 h after training, in Context A. Testing lasted for 10 min; freezing from min 2 through 6 was averaged to assess contextual fear expression [37,66] (Appendix A). The full time course of the testing period was evaluated to determine contextual fear expression and extinction. No shocks were administered during testing.

### 4.6. Swim Stress

Mice were moved to a holding room approximately 30 ft away from the swim stress testing room a minimum of 1 h prior to test commencement to acclimate. Control (“no swim”) mice were included in every cohort. These mice experienced a sham stressor, involving moving them to the holding room, acclimating, then being moved to individual transport cages during the ‘test’ period and put half-on a heating pad. Mice that did undergo a swim stress test were, after the acclimation period, brought in an individual transport cage directly to the swim stress testing room and immediately (and gently) placed in a tank of water (26 cm radius × 36.8 cm high) that was between 22.5 and 24.0 °C. This swim stress lasted for 6 min, and the entirety was recorded with a digital video camera for offline hand scoring of behaviors (Solomon Coder v. beta 19.08.02; https://solomon.andraspeter.com/, accessed on 16 November 2023). Fresh water was used for every single mouse, and the tank was rinsed thoroughly between each mouse. An experimenter, remaining silent and still, watched each entire swim in real time to ensure that no mouse was ever at risk of becoming submerged below the water’s surface. At the test end, mice were immediately (and gently) removed from the water, hand-dried with clean paper towels, and then placed in an individual transport cage half-on a heating pad. Mice remained half-on heating pads in their individual transport cages for at least 15 min, or until their fur was completely dry, whichever came second.

### 4.7. Study Phases

Two phases were conducted for this study, each with separate mice (Figure 1). The numbers of mice within each subgroup (Phase, fear conditioning type and stage, swim vs. no-swim, sex, genotype) range between n = 7 and 17; specific numbers for each subgroup in Figure 2, Figure 3, Figure 4, Figure 5, Figure 6, Figure 7, Figure 8, Figure 9, Figure 10 and Figure 11 are detailed in Appendix A. Phase 1 involved mice first undergoing context or cued fear conditioning, followed 4 weeks after the last fear test by swim stress. Phase 2 was the reverse, with mice first undergoing a swim stress, then 4 weeks later commencing either cued or context fear conditioning. No-swim mice were used as controls in both Phases 1 and 2. All mice underwent fear conditioning, because we have previously published on swim stress behavior in the absence of fear conditioning or any other stressor [10]. This approach, combined with our within-subjects design for each phase, was to minimize the number of mice used in accordance with the three Rs [67].

### 4.8. Tissue Collection

Tissue was collected 2 h after swim stress (or placement half-on heating pad, for no-swim controls) for Phase 1 and 2 h after the final fear test for Phase 2 (Figure 1). Previously, we observed no differences in serum corticosterone levels 30 min after swim stress (see Appendix A), the time point at which corticosterone peaks following an acute stressor. Given this information, plus our experimental design of heterotypic stressors spaced 4 weeks apart, we intentionally evaluated corticosterone levels 2 h after the last behavioral test for each phase. This allowed us to determine if the descending limb of the corticosterone curve was impacted by PMAT deficiency, biological sex, stressor history, or any interaction thereof. Just prior to tissue collection, mice were briefly anesthetized with isoflurane, then rapidly decapitated to obtain trunk blood. Ears were also collected at this time for reverification of genotype. Blood was allowed to clot at room temperature (20 ± 2 °C) for 30 min, then it was spun in a tabletop centrifuge at 3500 rpm and 4 °C for 30 min. Serum supernate was collected and placed in a clean tube, then serum and ears were frozen and stored at −80 °C until analyses. Serum corticosterone levels were quantified using corticosterone ELISA kits (ADI-900-097, Enzo Life Sciences, Inc., Farmingdale, NY, USA). Log transformation of serum corticosterone levels was performed prior to analyses to correct for the typical skewness of these data [38,39,40].

### 4.9. Data Graphing & Statistical Analyses

Data were graphed using GraphPad Prism (v 10.0.2 (171); GraphPad Software, San Diego, CA, USA), showing the mean ±95% confidence interval (CI), plus individual data points when not showing repeated measures data. Data were analyzed with GraphPad Prism and IBM SPSS Statistics (v 29.0.1.0 (171), IBM, Armonk, NY, USA). Significance thresholds were set a priori at *p* < 0.05, and non-significant trends (*p* < 0.10) were only examined if their corresponding partial η^2^ > 0.060. Analyses were performed within each phase and each form of fear conditioning (e.g., Phase 1 cued, Phase 2 context, etc.). Repeated measures data were analyzed within each training/testing stage and within each sex, using 3-way repeated-measures ANOVAs (time × PMAT genotype × swim condition) and pairwise comparisons with Bonferroni correction, or 2-way repeated-measures ANOVAs (PMAT genotype × time) and Holm–Šídák post hoc testing. Greenhouse–Geisser corrections were employed for within-subjects analyses. Average contextual fear expression (minutes 2 through 6) was analyzed in Phase 1 by a 2-way ANOVA (PMAT genotype × sex; because no effects of swim were detected, so data were collapsed across swim condition), and in Phase 2 by a 3-way ANOVA (PMAT genotype × sex × swim condition), all with Holm–Šídák post hoc tests (Appendix A). Measurements of serum corticosterone were analyzed within each phase and form of fear conditioning (cued or context) by a 3-way ANOVA (PMAT genotype × sex × swim condition) and Holm–Šídák post hoc tests. Swim measures were analyzed by a 2-way ANOVA (PMAT genotype × sex) within each phase and form of fear conditioning (cued or context) and pairwise comparisons with Bonferroni correction. Some data loss occurred for the following reasons: software malfunctions (e.g., file did not save); equipment malfunction (e.g., camera was not displaying real-time images); operator error (e.g., chamber door left open by the experimenter, and mouse departed chamber). Additionally, some mouse behavior indicated impairments in fear learning or excessive unconditioned fear. Exclusion criteria were as follows: (1) freezing > 75% in any 5 (context) or 30 (cued) s bin prior to the first mild foot shock being administered; or (2) freezing < 25% for the first five tones (first stage of cued fear testing in Context B), for every 30 s bin of testing (context fear testing), or for all five tones of cued fear testing in Context A (i.e., cued fear renewal). Specific details of all instances are in the Appendix A. The criterion to exclude outliers was a priori assigned as >5 standard deviations ± mean.

## Figures and Tables

**Figure 1 ijms-24-16494-f001:**
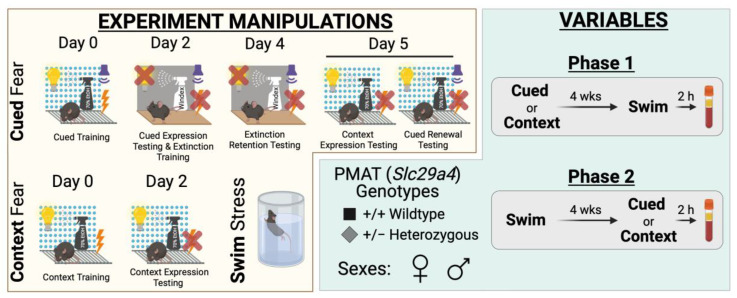
Experimental manipulations and variables for study. Experimental manipulations involved cued fear conditioning, context fear conditioning, and swim stress (yellow compartment, left side). Cued fear conditioning (**top**, yellow compartment) involved cued fear training in Context A (visible light, grid floor, patterned background, ethanol scent) on Day 0, followed by cued fear expression testing and cued extinction training on Day 2 in Context B (infrared light, smooth floor, no background, Windex scent). On Day 4, testing of extinction retention occurred in Context B, then Day 5 involved testing mice in Context A for context expression testing followed immediately by cued fear renewal testing. Note that for Day 5, the graphic shows the different conditions for context expression testing (**left**; 10 min) and cued fear renewal testing (**right**; 5 min) for clarity, but that in practice these tests occurred within the same continuous 15 min testing session. Context fear conditioning (bottom left, yellow compartment) involved context fear training in Context A on Day 0, with testing occurring in Context A on Day 2. Swim stress involved a 6 min inescapable immersion in room temperature water (**bottom** right, yellow compartment). Variables involved plasma membrane monoamine transporter (PMAT, *Slc29a4*) genotype, sex, swim condition, and the timeline of stressor exposure (Phase 1 or 2) (green compartment, right side). Wild-type (+/+) or heterozygous (+/−) mice of both sexes were used (bottom left, green compartment). Phase 1 (top right, green compartment) involved exposing mice to either cued or context fear conditioning, followed 4 weeks later by swim stress; 2 h after swim stress, blood was collected for serum corticosterone analyses. Phase 2 (bottom right, green compartment) had mice undergo swim stress, followed 4 weeks later by either cued or context fear conditioning; 2 h after the last test (Day 5, cued; Day 2, context), blood was collected for serum corticosterone analyses.

**Figure 2 ijms-24-16494-f002:**
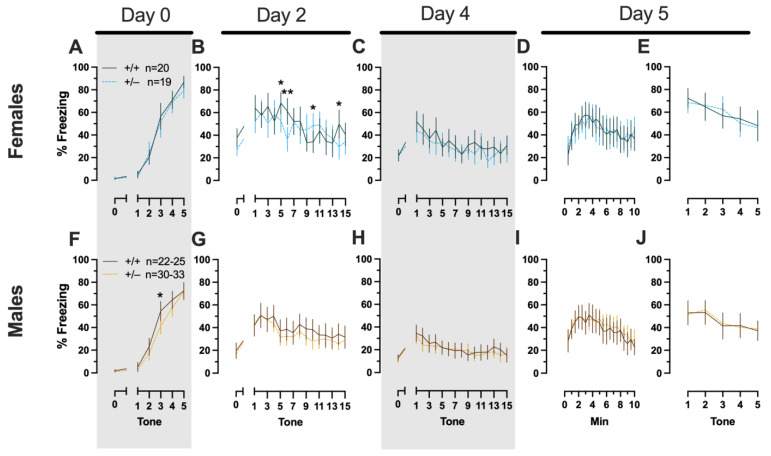
Phase 1 cued fear conditioning in female and male mice, collapsed across swim condition. Female (**A**–**E**) wild types are represented by teal solid lines, and female heterozygotes are represented by blue dashed lines. Male (**F**–**J**) wild types are represented by orange solid lines, and male heterozygotes are represented by yellow dashed lines. Fear conditioning commenced 4 weeks prior to swim stress exposure for Phase 1. Mice were trained on Day 0 in Context A, then two days later (Day 2), mice were placed in Context B for cued fear expression testing as well as cued fear extinction training. Two days thereafter, mice underwent the identical procedure for the purposes of testing extinction retention (Day 4; **C**,**H**). One day later (Day 5), mice were placed in Context A and were tested for context fear expression (**D**,**I**) followed immediately by testing cued fear renewal (**E**,**J**). Data are percent time spent freezing for each 30 s period indicated. Data were analyzed within each training/testing stage and within each sex using two-way repeated-measures ANOVAs (PMAT genotype × time) and Holm–Šídák post hoc testing and are graphed as mean ± 95% confidence interval. * *p* = 0.027, * *p* = 0.042, * *p* = 0.016 (left to right, panel **B**); * *p* = 0.021 (panel **F**); ** *p* = 0.002 (panel **B**); indicate the difference between heterozygous and wild type within the same sex at the indicated timepoint.

**Figure 3 ijms-24-16494-f003:**
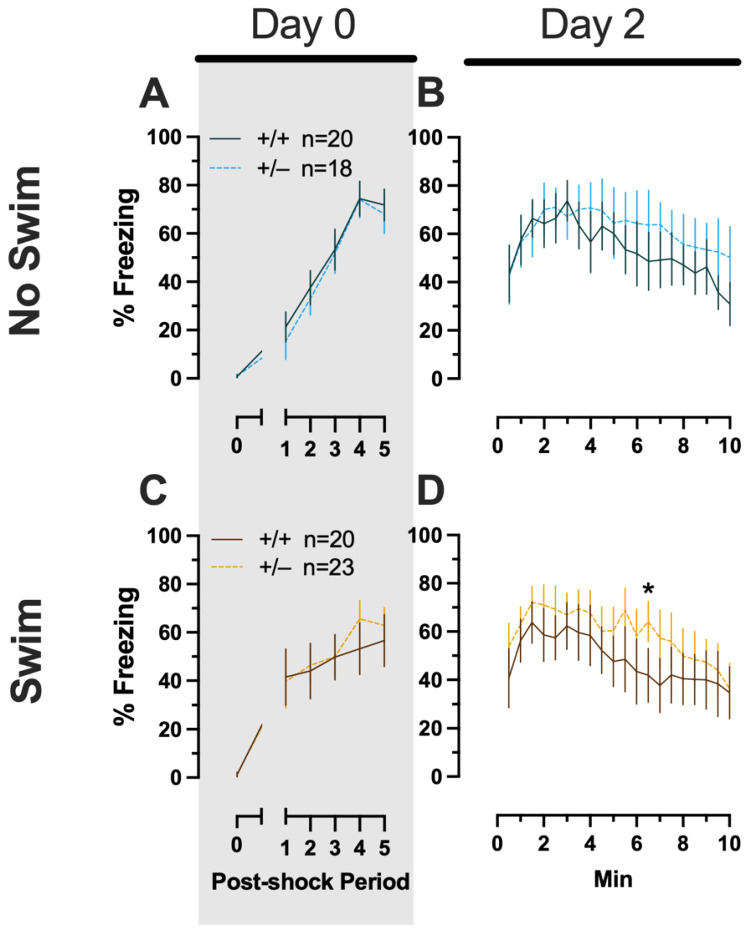
Phase 1 context fear conditioning in female and male mice, collapsed across swim condition. Female (**A**,**B**) wild types are represented by teal solid lines, and female heterozygotes are represented by blue dashed lines. Male (**C**,**D**) wild types are represented by orange solid lines, and male heterozygotes are represented by yellow dashed lines. Fear conditioning commenced 4 weeks prior to swim stress exposure for Phase 1. Mice were trained on Day 0 in Context A (**A**,**C**). Two days later (Day 2), mice were placed back in Context A to test for context fear expression (**B**,**D**). Data are percent time spent freezing for each 30 s period following the foot shock (**A**,**C**) or every 30 s of testing (**B**,**D**). Data were analyzed within each training/testing stage and within each sex using two-way repeated-measures ANOVAs (PMAT genotype × time) and Holm–Šídák post hoc testing and are graphed as mean ± 95% confidence interval. * *p* = 0.039 indicates a difference between heterozygous and wild type within the same sex at the indicated time point.

**Figure 4 ijms-24-16494-f004:**
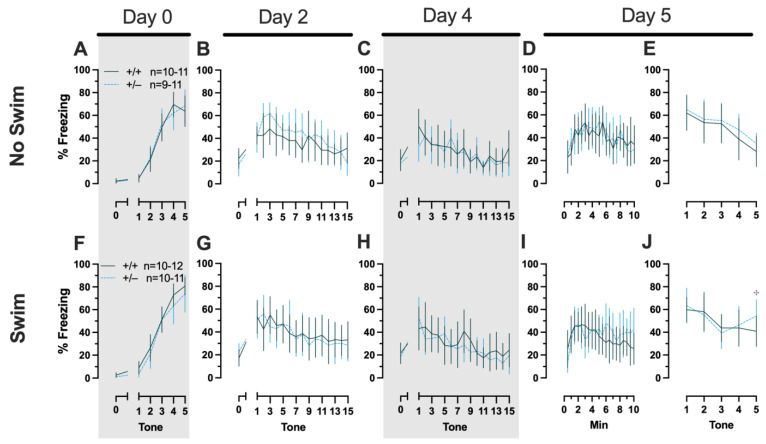
Phase 2 cued fear conditioning in female mice, separated by swim condition. Female wild types are represented by teal solid lines, and female heterozygotes are represented by blue dashed lines. Fear conditioning occurred 4 weeks after swim stress exposure for Phase 2. Mice were trained on Day 0 in Context A (**A**,**F**). Two days later (Day 2), mice were placed in Context B (**B**,**G**); this served as cued fear expression testing as well as cued fear extinction training. Two days thereafter, mice underwent the identical procedure for the purposes of testing extinction retention (Day 4; **C**,**H**). One day later (Day 5), mice were placed in Context A to test for context fear expression (**D**,**I**) then immediately thereafter tested for cued fear renewal (**E**,**J**). Data are percent time spent freezing for each 30 s period indicated. Data were analyzed within each training/testing stage and within each sex, using 3-way repeated-measures ANOVAs (time × PMAT genotype × swim condition) and pairwise comparisons with Bonferroni correction, and are graphed as mean ± 95% confidence interval. **^✣^** indicates *p* = 0.021 difference between no-swim and swim conditions within the same sex and genotype (heterozygous) at the indicated time point.

**Figure 5 ijms-24-16494-f005:**
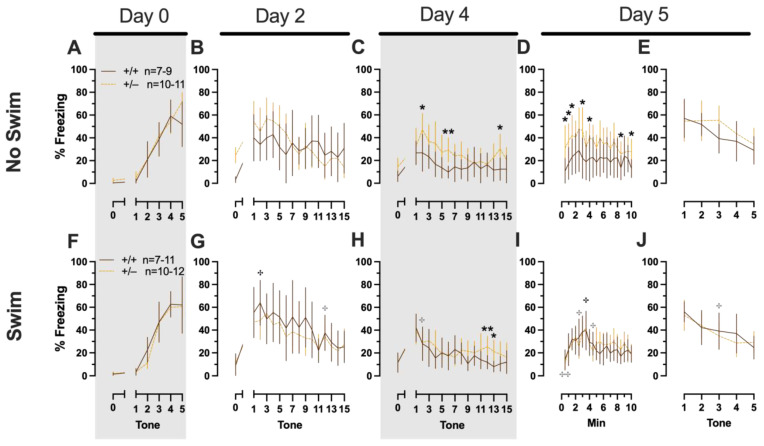
Phase 2 cued fear conditioning in male mice, separated by swim condition. Male wild types are represented by orange solid lines, and male heterozygotes are represented by yellow dashed lines. Fear conditioning occurred 4 weeks after swim stress exposure for Phase 2. Mice were trained on Day 0 in Context A (**A**,**F**), then two days later (Day 2) placed in Context B (**B**,**G**), where they underwent cued fear expression testing as well as cued fear extinction training. Two days thereafter, mice underwent the identical procedure for the purposes of testing extinction retention (Day 4; **C**,**H**). One day later (Day 5), mice were placed in Context A for context fear expression (**D**,**I**). Immediately after context fear expression testing, mice were tested for cued fear renewal (**E**,**J**). Data are percent time spent freezing for each 30 s period indicated. Data were analyzed within each training/testing stage and within each sex, using 3-way repeated-measures ANOVAs (time × PMAT genotype × swim condition) and pairwise comparisons with Bonferroni correction, and are graphed as mean ± 95% confidence interval. * *p* = 0.023, * *p* = 0.015 (left to right, panel **C**); * *p* = 0.021, * *p* = 0.029,* *p* = 0.026, * *p* = 0.023, * *p* = 0.043, * *p* = 0.046, * *p* = 0.015 (left to right, panel **D**); * *p* = 0.039 (panel **H**); ** *p* = 0.002 (panel **C**); ** *p* = 0.008 (panel **H**) indicate difference between heterozygous and wild type within the same sex at the indicated time point. ^**✣**^ *p* = 0.025, ^✣^ *p* = 0.019 (left to right, panel **G**); **^✣^** *p* = 0.031 (panel **H**); **^✣^** *p* = 0.019, **^✣^** *p* = 0.035, **^✣^** *p* = 0.013 (left to right, panel **I**); **^✣✣^** *p* = 0.009 (panel **I**); **^✣^** *p* = 0.015 (panel **J**) indicates difference between no-swim and swim conditions within the same sex and genotype (indicated by color; black for wild-type, grey for heterozygous) at the indicated time point.

**Figure 6 ijms-24-16494-f006:**
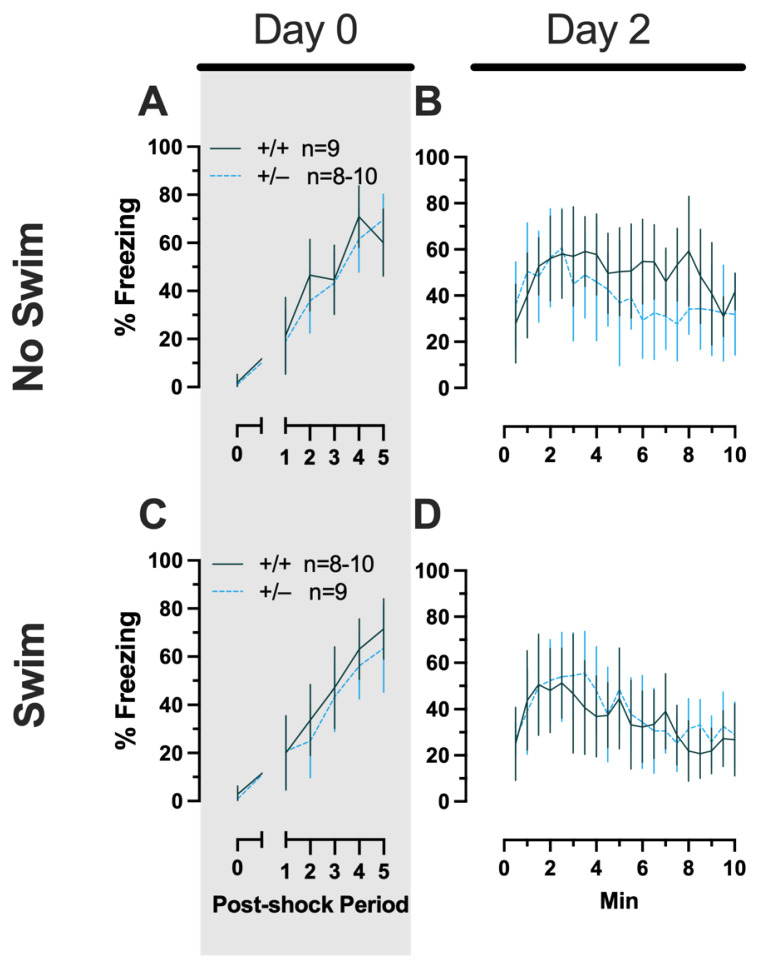
Phase 2 context fear conditioning in female mice, separated by swim condition. Female wild types are represented by teal solid lines, and female heterozygotes are represented by blue dashed lines. Fear conditioning occurred 4 weeks after swim stress exposure for Phase 2. Mice were trained on Day 0 in Context A (**A**,**C**). Two days later (Day 2), mice were placed back in Context A to test for context fear expression (**B**,**D**). Data were analyzed within each training/testing stage and within each sex, using 3-way repeated-measures ANOVAs (time × PMAT genotype × swim condition) and pairwise comparisons with Bonferroni correction. Data are percent time spent freezing for each 30 s period following the foot shock (**A**,**C**) or every 30 s of testing (**B**,**D**). Data are graphed as mean ± 95% confidence interval.

**Figure 7 ijms-24-16494-f007:**
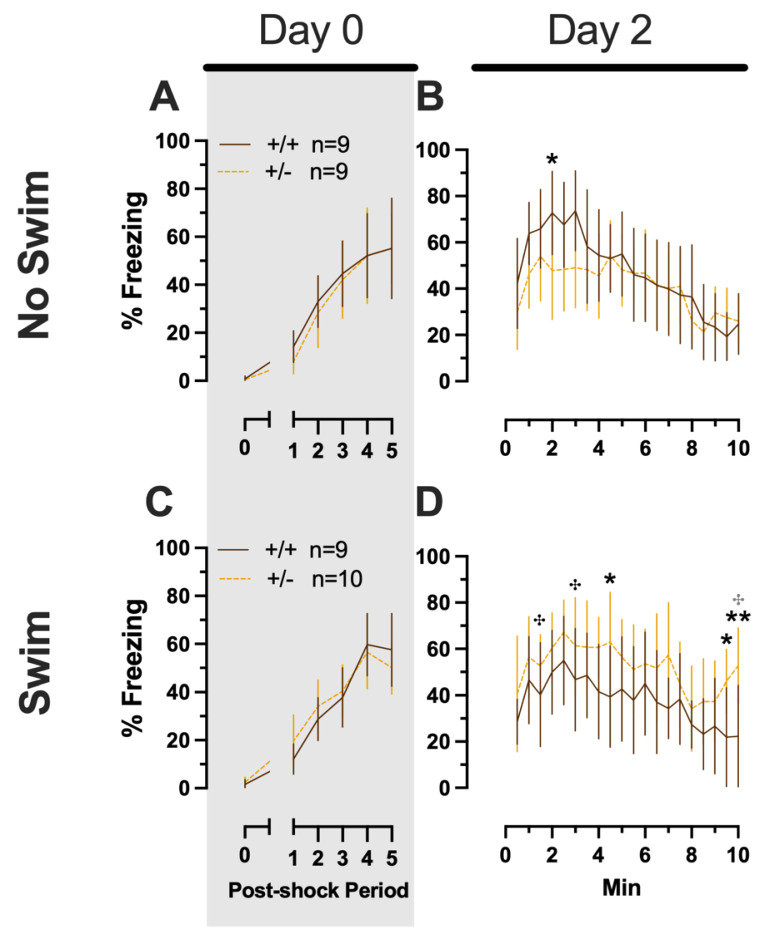
Phase 2 context fear conditioning in male mice, separated by swim condition. Male wild types are represented by orange solid lines, and male heterozygotes are represented by yellow dashed lines. Fear conditioning occurred 4 weeks after swim stress exposure for Phase 2. Mice were trained on Day 0 in Context A (**A**,**C**). Two days later (Day 2), mice were placed back in Context A to test for context fear expression (**B**,**D**). Data are percent time spent freezing for each 30 s period following the foot shock (**A**,**C**) or every 30 s of testing (**B**,**D**). Data were analyzed within each training/testing stage and within each sex, using 3-way repeated-measures ANOVAs (time × PMAT genotype × swim condition) and pairwise comparisons with Bonferroni correction, and are graphed as mean ± 95% confidence interval. * *p* = 0.034 (panel **B**); * *p* = 0.048, * *p* = 0.013 (left to right, panel **D**); ** *p* = 0.004 (panel **D**) indicate difference between heterozygous and wild type within the same sex at the indicated time point. **^✣^** *p* = 0.030, **^✣^** *p* = 0.035, **^✣^** *p* = 0.011 (left to right, panel **D**) indicate difference between no-swim and swim conditions within the same sex and genotype (indicated by color; black for wild-type, grey for heterozygous) at the indicated time point.

**Figure 8 ijms-24-16494-f008:**
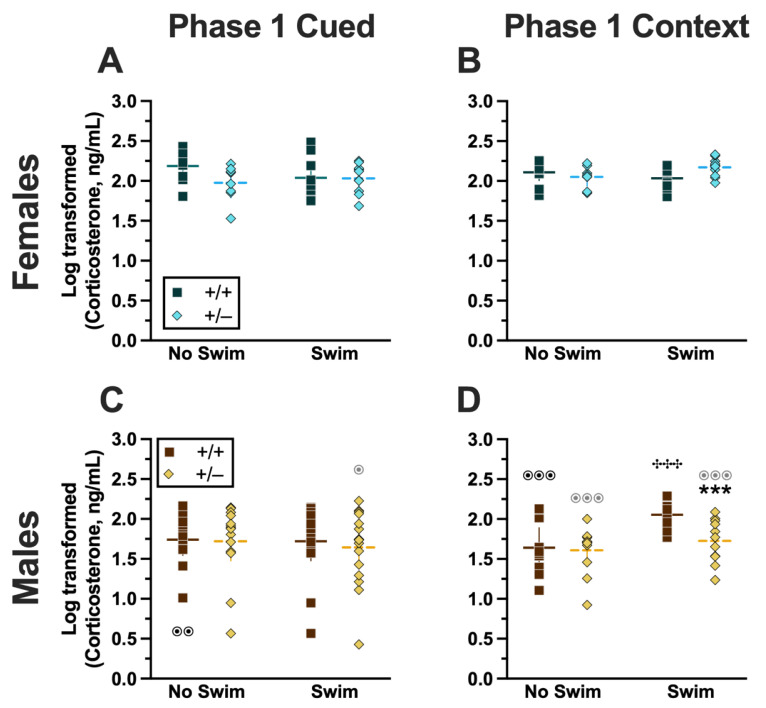
Log-transformed corticosterone levels in mice from Phase 1. Mice in Phase 1 had blood collected 2 h following swim stress to measure serum corticosterone levels. Female (**A**,**B**) wild types are represented by teal squares, and female heterozygotes are represented by blue diamonds. Male (**C**,**D**) wild types are represented by orange squares, and male heterozygotes are represented by yellow diamonds. Phase 1 cued data (**A**,**C**) and Phase 1 context data (**B**,**D**) are graphed in columns separated by sex. Data are log-transformed serum corticosterone levels, showing individual data points. Data were analyzed within each Phase and form of fear conditioning (cued or context) by a 3-way ANOVA (PMAT genotype × sex × swim condition) and Holm–Šídák post hoc tests. Horizontal lines are shown as the mean, with vertical lines as ±95% confidence interval. *** *p* = 0.001 (panel **D**) indicates difference between heterozygous and wild type within the same sex and same swim condition. **^✣✣✣^** *p* < 0.001 (panel **D**) indicates difference between no-swim and swim conditions within the same sex and genotype. ^⊙⊙^ *p* = 0.008, ^⊙^ *p* = 0.014 (left to right, panel **C**); ^⊙⊙⊙^ *p* < 0.001, ^⊙⊙⊙^ *p* < 0.001, ^⊙⊙⊙^ *p* < 0.001 (left to right, panel **D**) indicate difference between sexes within the same genotype and swim condition.

**Figure 9 ijms-24-16494-f009:**
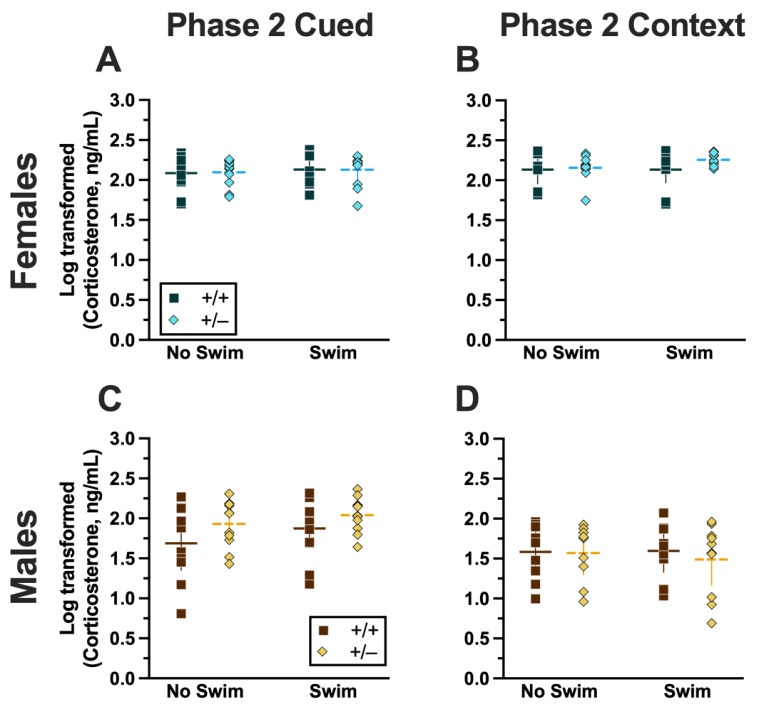
Log-transformed corticosterone levels in mice from Phase 2. Mice in Phase 2 had blood collected 2 h following context fear testing and cued fear renewal to measure serum corticosterone levels. Female (**A**,**B**) wild types are represented by teal squares, and female heterozygotes are represented by blue diamonds. Male (**C**,**D**) wild types are represented by orange squares, and male heterozygotes are represented by yellow diamonds. Phase 2 cued data (**A**,**C**) and Phase 2 context data (**B**,**D**) are graphed in columns separated by sex. Data are log-transformed serum corticosterone levels, showing individual data points. Data were analyzed within each phase and form of fear conditioning (cued or context) by a 3-way ANOVA (PMAT genotype × sex × swim condition). Horizontal lines are shown as the mean, with vertical lines as ±95% confidence interval.

**Figure 10 ijms-24-16494-f010:**
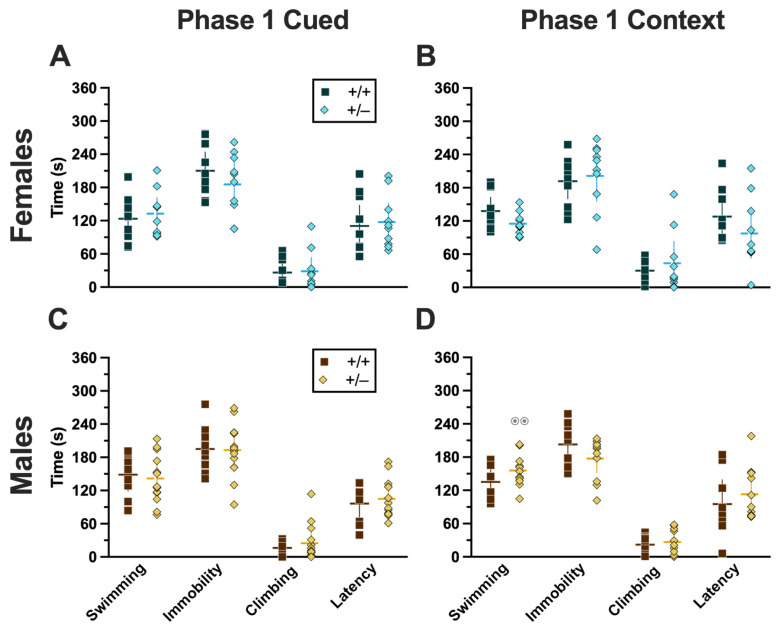
Behaviors during the swim stressor in Phase 1 mice. Mice in Phase 1 assigned to swim stress experienced an acute 6 min swim stressor 4 weeks after undergoing Cued (**A**,**C**) or Context (**B**,**D**) fear conditioning. Female (**A**,**B**) wild types are represented by teal squares, and female heterozygotes are represented by blue diamonds. Male (**C**,**D**) wild types are represented by orange squares, and male heterozygotes are represented by yellow diamonds. Data are the amount of time in seconds spent swimming, immobile, or climbing; or the amount of time until the first bout of immobility (i.e., latency). Data were analyzed by a 2-way ANOVA (PMAT genotype × sex) within each phase and form of fear conditioning (cued or context) and pairwise comparisons with Bonferroni correction. Horizontal lines are shown as the mean, with vertical lines as ±95% confidence interval. ^⊙⊙^ *p* = 0.003 (panel **D**) indicates difference between sexes within the same genotype and swim behavior.

**Figure 11 ijms-24-16494-f011:**
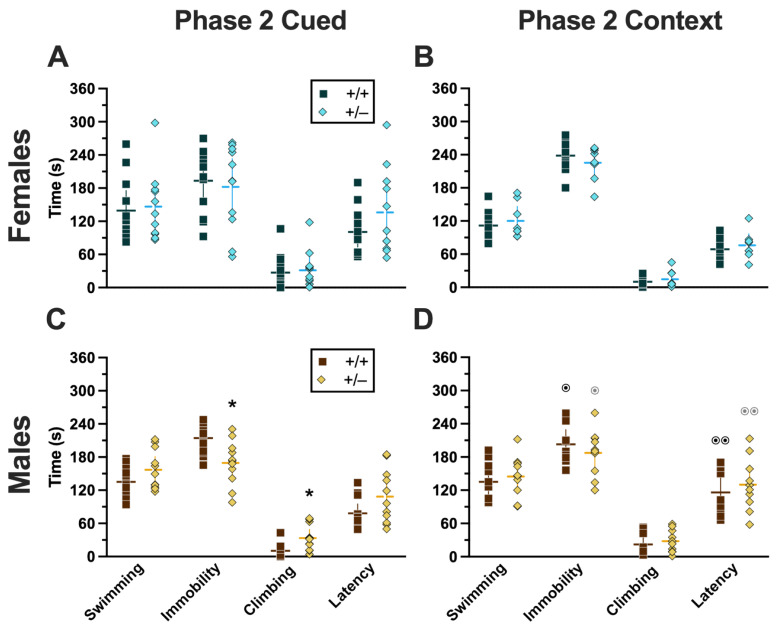
Behaviors during the swim stressor in Phase 2 mice. Mice in Phase 2 assigned to swim stress experienced an acute 6 min swim stressor 4 weeks before undergoing Cued (**A**,**C**) or Context (**B**,**D**) fear conditioning. Female (**A**,**B**) wild types are represented by teal squares, and female heterozygotes are represented by blue diamonds. Male (**C**,**D**) wild types are represented by orange squares, and male heterozygotes are represented by yellow diamonds. Data are the amount of time in seconds spent swimming, immobile, or climbing; or the amount of time until the first bout of immobility (i.e., latency). Data were analyzed by a 2-way ANOVA (PMAT genotype × sex) within each phase and form of fear conditioning (cued or context) and pairwise comparisons with Bonferroni correction. * *p* = 0.049, * *p* = 0.043 (left to right, panel **C**) indicate difference between heterozygous and wild type within the same sex for the same swim behavior. ^⊙^ *p* = 0.032, ^⊙^ *p* = 0.030, ^⊙⊙^ *p* = 0.006, ^⊙⊙^ *p* = 0.003 (left to right, panel **D**) indicate difference between sexes within the same genotype and swim behavior.

## Data Availability

All data will be made publicly available at the time of publication through OSF: https://osf.io/qwev3/?view_only=2426baca6f634d06a5fb40100573c54b, accessed on 16 November 2023.

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
