# Peer review of "Heterotypic Stressors Unmask Behavioral Influences of PMAT Deficiency in Mice"

_ijms, 2023, doi:10.3390/ijms242216494_

Round 1
Reviewer 1 Report
Comments and Suggestions for Authors
In this manuscript, Weber et al. investigated, with extensive in vivo experiments, how reduced PMAT function affects response to stressors in homo- and heterozygous mice of both sexes.
In the introduction you refer to uptake-1 and uptake-2 - this terminology was mostly used at a time when the high and low capacity monoamine transporters have not been thoroughly identified yet. Now it is more appropriate to distinguish SLC6 and SLC22 or SLC29 monoamine transporters via their differences in affinity and capacity for substrate transport.
The decision to use heterozygous mice (and not KOs) to mimic reality is sound but I worry that expression levels in heterozygous mice might still be quite high - did you compare expression in tissues of WT and heterozygous mice? If not, I think that this is a limitation that needs to be stated.
Was a power analysis conducted and how were the animal numbers determined?
Could you explicitly state the rationale for including the measurement of corticosterone levels? Did you measure it as a surrogate parameter for stress occurrence? I wonder because it has been shown that corticosterone barely interacts with PMAT (doi: 10.1002/cpt.442).
How do you explain that in Phase 1 cued, time x genotype were significant in females and not males, whereas in Phase 1 context time and genotype were sperately significant only in males? Is there an argument for why the interaction term was not significant?
The data shown for corticosterone is very interesting but also quite hard to interpret concerning the large differences between conditions. The discussion would benefit from an even more in-depth explanation/discussion of this phenomenon.
In general, I do not think that all ANOVA results need to be shown as a table (in the main text) but rather only the significant factors described. As is, the manuscript is sometimes hard to follow because the tables make up a large part of it and are not always immediately referred to.
The work is massive and shows some interesting results that are, unfortunately, as you also state in the discussion ("Adding to the confusion [...]") quite hard to interpret. I think that the discussion would benefit from less speculation and rather an analysis that points at trends but also inconsistencies in the data to develop future hypotheses to look into.
The data was rigoursly gathered and honestly reported. The results and discussion sections, with the many different aspects being reported, would benefit from some more structure to make them easier to digest for the reader.
Reviewer 2 Report
Comments and Suggestions for Authors
There is no doubt that the research results presented in manuscript ijms-2627478 are innovative. They concern the response to stress in female and male mice, depending on the function of PMAT (plasma membrane monoamine transporter). However, in my opinion, the big problem is the ill-considered way of presenting the results.
Critical remarks:
Both the title of the manuscript and the abstract should be informative. There is no information in the title of the work or in the abstract that the research was conducted on an experimental model of stress in mice.
In the "Methods" section, the characteristics of the experimental animals do not provide group sizes. This information is only included in the explanations below the figures.
The description of the results obtained is very confusing.
Authors should reconsider the way they present data. In my opinion, Tables 1-13 in their current form are appropriate for presentation in supplemental materials rather than in the main text of the manuscript.
The captions for Figures 2, 4, 5 and 8 are also unclear and too long.
According to the standards of scientific publications and the principles of interpretation of results in terms of statistical significance, statistically significant differences occur at P<0.05. And only such should be marked on the charts (in line 605, *p=0.43 is therefore redundant).
For clarity of interpretation of the obtained results, please verify the graphical indications of statistical significance, according to the rule: one symbol at P<0.05, two symbols at P<0.01 and three symbols at P<0.001. I propose to adopt such a scheme for registering statistical significance in figure captions. Symbols in figures denoting statistical significance should be in a different color than the data presented (points/lines, as appropriate).
The article requires editorial correction. You should, among others: pay attention to the spaces between words (there are not single spaces everywhere, e.g. lines 10, 11, 16, 22, 23, 30, 42, 68, 115, 131, 134, 205, 207).
Line 9 - should probably be "selected" instead of "select".
Line 407 - unnecessary dot before "Data".
In addition, the record of references also requires verification. Among other things, their order numbers were doubled. The authors also do not use a unified way of writing abbreviations of journal titles (dots/no dots).
My recommendation: RECONSIDER AFTER MAJOR REVISION

Round 2
Reviewer 2 Report
Comments and Suggestions for Authors
The authors significantly improved the manuscript by following the reviewers' recommendations. Changing the way the results are presented definitely makes them easier to interpret. However, I still have reservations about the way statistically significant differences are presented, both in the figures and in the captions below them.
In the captions under the figures, the authors unnecessarily provide the exact p value. In some cases, the p values are very close to each other (e.g. caption under Figure 2, line 305; caption under Figure 5, line 411). Therefore, information given in the form p < 0.05, p < 0.01 or p < 0.001 is sufficient.
The captions can include general information that one symbol means p<0.05, two - p<0.01 and three - p<0.001. This information will replace specifying symbols for each p.
In figures, symbols denoting statistical significance should be of the same size (asterisks and circles).
Moreover, it is good practice to provide the types of statistical tests used for the analysis in the captions under the figures.
Figures 2-11: the lines or points presenting the results of the "+/+" and "+/-" groups are too close in color. The use of contrasting colors in the figures will significantly improve the visualization of the results.
The text of the manuscript still requires editorial correction. Among other things, you should pay attention to the principle of single space between words.
